# Cystathionine Gamma Lyase Is Regulated by Flow and Controls Smooth Muscle Migration in Human Saphenous Vein

**DOI:** 10.3390/antiox12091731

**Published:** 2023-09-07

**Authors:** Shuang Zhao, Céline Deslarzes-Dubuis, Severine Urfer, Martine Lambelet, Sébastien Déglise, Florent Allagnat

**Affiliations:** Department of Vascular Surgery, Lausanne University Hospital, 1005 Lausanne, Switzerland; shuang.zhao@chuv.ch (S.Z.); celine.deslarzes@chuv.ch (C.D.-D.); sev.urfer@bluewin.ch (S.U.); martine.lambelet@chuv.ch (M.L.); sebastien.deglise@chuv.ch (S.D.)

**Keywords:** hydrogen sulfide, cystathionine-γ-lyase, CSE, H_2_S, intimal hyperplasia, venous bypass

## Abstract

The saphenous vein is the conduit of choice for bypass grafting. Unfortunately, the hemodynamic stress associated with the arterial environment of the bypass vein graft leads to the development of intimal hyperplasia (IH), an excessive cellular growth and collagen deposition that results in restenosis and secondary graft occlusion. Hydrogen sulfide (H_2_S) is a ubiquitous redox-modifying gasotransmitter that inhibits IH. H_2_S is produced via the reverse trans-sulfuration pathway by three enzymes: cystathionine γ-lyase (CSE), cystathionine β-synthase (CBS) and 3-mercaptopyruvate sulfurtransferase (3-MST). However, the expression and regulation of these enzymes in the human vasculature remains unclear. Here, we investigated the expression of CSE, CBS and 3-MST in segments of native human saphenous vein and large arteries. Furthermore, we evaluated the regulation of these enzymes in vein segments cultured under static, venous (7 mmHg pressure) or arterial (100 mmHg pressure) pressure. CSE was expressed in the media, neointima and intima of the vessels and was negatively regulated by arterial shear stress. Adenoviral-mediated CSE overexpression or RNA interference-mediated CSE knock-down revealed that CSE inhibited primary human VSMC migration but not proliferation. We propose that high shear stress in arteriovenous bypass grafts inhibits CSE expression in both the media and endothelium, which may contribute to increased VSMC migration in the context of IH.

## 1. Introduction

Arterio-venous bypass surgery is one of the main approaches for revascularization of chronic limb-threatening ischemia (CLTI) patients. However, veins are not designed to support arterial pressure and undergo significant vascular remodeling to adapt to the arterial environment. This remodeling is accompanied by the development of intimal hyperplasia (IH), i.e., the formation of a collagen-rich neointima layer between the media and the innermost layer (intima/endothelium) of the vein. IH is due to a cascade of cellular events leading to the differentiation, proliferation and migration of vascular smooth muscle cells (VSMC) from the vessel wall into the intima [1]. This excessive cell growth and collagen deposition eventually lead to reduced blood flow (restenosis) or occlusion of the bypass. Approximately 30 to 50% of the saphenous grafts fail 1–18 months after implantation [2].

Hydrogen sulfide (H_2_S) contributes to the homeostasis of a wide range of systems, including the cardiovascular system [3]. Notably, endogenous H_2_S bioavailability is attenuated in patients with CLTI and in patients with diabetes-related vascular inflammation [4]. Circulating H_2_S is also reduced in humans suffering from vascular occlusive disease [5,6], and patients undergoing surgical revascularization with lower H_2_S production capacity have higher postoperative mortality rates [7].

H_2_S is produced in mammalian cells through the reverse transulfuration pathway by two pyridoxal 5′-phosphate dependent enzymes, cystathionine γ-lyase (CSE) and cystathionine β-synthase (CBS), and by a combination of two additional enzymes, 3-mercaptopyruvate sulfurtransferase (3-MST) and cysteine aminotransferase (CAT). Mice lacking Cse display increased IH in a model of carotid artery ligation [8,9]. On the contrary, Cse overexpression decreases IH formation in a murine model of vein graft by carotid-interposition cuff technique [10]. In addition, we and others demonstrated that systemic treatment using diverse H_2_S donors inhibit IH in vivo in various models in rats [11], rabbits [12] and mice [8,9,13]. We also showed that several H_2_S donors inhibit IH ex vivo in human vein segments [9,13,14]. The study of Cse^−/−^ mice supports that CSE expression in endothelial cells (EC) is the main source of endogenous H_2_S production in vessels [15,16,17]. However, CSE expression has also been found in VMSC, and may contribute to VSMC proliferation and migration, vascular remodeling and IH [8,18]. CBS is also found in the cardiovascular system, but its role and distribution in vessels is unclear [3]. Other reports suggest a key role of 3-MST in H_2_S production in the vascular endothelium [19]. It was recently demonstrated that Cse expression is negatively regulated by shear stress in vitro [20]. This is in line with a previous study showing that only disturbed flow regions show discernible CSE protein expression after carotid artery ligation in the mouse [21]. However, the expression of CSE in human vessels remains poorly characterized. In this study, we studied the expression and regulation of CSE, CBS and 3MST in segments from healthy human saphenous vein and artery. We observed that CSE is expressed both in the endothelium and media of large vessels, whereas CBS expression is detectable only in the media, and 3-MST expression is mainly restricted to the endothelium of small vessels. Our data confirms that CSE expression in veins is negatively regulated by shear stress and, as a result, upregulated in absence of flow and downregulated in vein segments placed under arterial perfusion. We further confirm that CSE is involved in primary human VSMC migration, but not proliferation.

## 2. Materials and Methods

For details on materials and reagents please see the Appendix A.

### 2.1. Human Vessels Culture

Healthy human artery segments were obtained from patients undergoing vascular reconstruction as part of our biobank. Nine artery segments were used in this study. Healthy human saphenous vein segments were surplus segments of non-varicose veins from donors who underwent lower limb bypass surgery. Static vein culture was performed as described [13,14,22]. Briefly, segments of great saphenous vein were cut in 5 mm segments randomly distributed between conditions. One segment (D0) was immediately preserved in formalin or flash frozen in liquid nitrogen and the others were maintained in culture for 7 days in RPMI-1640 Glutamax I (Thermo Fisher Scientific, Ecublens, Switzerland supplemented with 10% FBS (Thermo Fisher Scientific, Ecublens, Switzerland) and 1% antibiotic solution (10,000 U/mL penicillin G, 10,000 U/mL streptomycin sulphate; Sigma-Aldrich; Merck KGaA, Darmstadt, Germany) in cell culture incubator at 37 °C, 5% CO_2_ and 21% O_2_.

Pulsatile vein culture using an ex vivo vein perfusion system was performed as previously described [23,24,25,26]. Upon harvest, veins were stored at 4 °C in a RPMI-1640 Glutamax medium, supplemented with 12.5% fetal calf serum (Thermo Fisher Scientific, Ecublens, Switzerland). Within 1 h after the surgery, the segments with an external diameter of 2.5–4 mm were divided in two equal parts. One part was fixed in either formalin for immunohistochemistry or rapidly frozen in liquid nitrogen for molecular analyses. A second part was perfused in the EVPS for 7 days to a pulsatile biphasic flow of 60 pulses/min under either low (LP = 7 mmHg; systolic/diastolic pressure = 8 ± 1/6 ± 1 mmHg) or high perfusion pressure (HP = 100 mmHg; systolic/diastolic pressure = 120 ± 5/90 ± 5 mmHg). Upon completion of the perfusion, the 5 mm proximal and distal ends, which attached the vein to the equipment, were discarded. A central 5 mm-thick ring was cut from the remaining segment and fixed in formalin for morphometry. The remaining fragments were frozen and reduced into powder for RT-PCR and Western blot analysis. The veins segments in the EVPS were maintained in RPMI-1640, supplemented with Glutamax, 12.5% fetal calf serum, 8% 70 kDa dextran and 1% antibiotic-antimycotic solution (10,000 U/mL penicillin G, 10 mg/mL streptomycin sulphate, 25 mg/mL amphotericin B, and 0.5 μg/mL gentamycin). This medium was changed every 2 days. In this study, eight veins obtained from randomly selected patients who underwent lower limb bypass surgery for critical ischemia were used.

### 2.2. Cell Culture

Human VSMCs were prepared from human saphenous vein segments as previously described [14,27]. Vein explants were plated on the dry surface of a cell culture plate coated with 1% Gelatine type B (Sigma-Aldrich; Merck KGaA, Darmstadt, Germany)). Explants were maintained in RPMI, 10% FBS medium in a cell culture incubator at 37 °C, 5% CO_2_, 5% O_2_ environment. Nine different veins/patients were used in this study to generate VSMC.

### 2.3. siRNA-Mediated Knock-Down and Adenoviral-Mediated Overexpression

CSE knockdown was performed using human siRNA targeting CTH (Ambion-Life Technologies, ID: s3710 and s3712). The control siRNA (siCtrl) was the AllStars Negative Control siRNA (QIAGEN AG, Hombrechtikon, Switzerland; SI03650318). VSMC grown at 70% confluence were transfected overnight with 30 nM siRNA using lipofectamin RNAiMax (Invitrogen, Thermo Fisher Scientific, Ecublens, Switzerland, 13778-075). After washing, cells were maintained in full media for 48 h prior to assessment.

CSE overexpression was achieved using a replication-deficient recombinant adenoviral (DE1/E3) vector. Adenoviral infection was achieved overnight in complete medium using AdCTH (kindly provided by James R. Mitchell [14], produced and purified by Vector Biolabs, Philadelphia, PA, USA), or the negative control virus Ad-eGFP (Vector Biolabs, Cat. No.: 1060). Adenoviral infection was conducted using the Adenovirus (CAR) receptor booster according to the manufacturer’s instructions (Cat:631470; Takara Bio Europe SAS, Saint-Germain-en-Laye, France). After washing, cells were maintained in full media for 48 h prior to assessment.

### 2.4. Histology

After 7 days in culture, or immediately upon vein isolation (D0), human vessel segments were fixed in buffered formalin, embedded in paraffin, cut into 5 µm sections and stained with VGEL as previously described [14]. Slides were scanned using a ZEISS Axioscan 7 Microscope Slide Scanner (ZEISS Microscopy AG, Basel, Switzerland. Polychrome Herovici staining was performed on paraffin sections as described [28]. Young collagen was stained blue, whereas mature collagen was pink. Cytoplasm was counterstained yellow. Hematoxylin was used to counterstain nuclei blue to black. For intimal and medial thickness, 96 (4 measurements/photos and 4 photos per cross section on six cross sections) measurements were performed. Two independent researchers blinded to the conditions did the morphometric measurements using the Olympus Stream Start 2.3 software (Olympus, Evident Europe GmbH, Basel, Switzerland) [13,14,22].

CSE, CBS and 3-MST immunohistochemistry were performed on paraffin sections. After rehydration and antigen retrieval (TRIS-EDTA buffer, pH 9, 1 min in an electric pressure cooker autocuiser Instant Pot duo 60 under high pressure), immunostaining was performed on human vein or artery sections using the EnVision^®+^ Dual Link System-HRP (DAB+) according to manufacturer’s instructions (Agilent Technologies (Schweiz) AG, Basel, Switzerland). Slides were further counterstained with hematoxylin. The positive immunostaining area was quantified using the Fiji (ImageJ 1.53t) software and normalized to the total area of the tissue by two independent observers blinded to the conditions.

### 2.5. Western Blotting

Vessels were flash-frozen in liquid nitrogen, grinded to power and resuspended in SDS lysis buffer (62.5 mM TRIS pH 6.8, 5% SDS, 10 mM EDTA). Protein concentration was determined by DC protein assay (Bio-Rad Laboratories, Reinach, Switzerland). Approximately 10 to 20 µg of protein were loaded per well. Primary cells were washed once with ice-cold PBS and directly lysed with Laemmli buffer as previously described [14]. Lysates were resolved by SDS-PAGE and transferred to a PVDF membrane (Immobilon-P, Millipore, Merck KGaA, Darmstadt, Germany). Immunoblot analyses were performed as previously described [27] using a CSE antibodies described in Appendix A. Membranes were revealed by enhanced chemiluminescence (Immobilon, Millipore) using the Azure 280 device (Azure Biosystems, Dublin, CA, USA) and analyzed using the Fiji (ImageJ 1.53t) software. Protein abundance was normalized to total protein using Pierce™ Reversible Protein Stain Kit for PVDF Membranes (cat 24585; Thermo Fisher Scientific).

### 2.6. Reverse Transcription and Quantitative Polymerase Chain Reaction (RT-qPCR)

Flash frozen vessels powder was homogenized in TriPure™ Isolation Reagent (Roche Diagnostics AG, Rotkreuz, Switzerland), and total RNA was extracted according to the manufacturer’s instructions. After RNA reverse transcription (Prime Script RT reagent, Takara Bio Europe SAS, Saint-Germain-en-Laye, France), cDNA levels were measured by qPCR Fast SYBR™ Green Master Mix (Ref: 4385618, Applied Biosystems, ThermoFischer Scientific AG, Switzerland) in a Quant Studio 5 Real-Time PCR System (Applied Biosystems, ThermoFischer Scientific AG, Switzerland), using the primers described in Appendix A.

### 2.7. Lead Acetate (CSE/CBS Activity Assay)

Flash frozen vessels powder was homogenized in passive lysis buffer (Promega) and protein content was determined using a BCA protein assay (ThermoFischer Scientific AG, Switzerland). The lead acetate assay was performed as previously described [14,17,29]. Briefly, using a 96 well plate, 300 μg of proteins were diluted into 100 µL of PBS supplemented with 10 mM Cysteine and 1 mM pyridoxal phosphate as substrate and cofactor for CSE and CBS. The plate was then covered with Whatman paper impregnated with 20 mM lead acetate and incubated at 37 °C for 5 h. Lead sulfide precipitate on the Whatman paper were scanned using a high-resolution scanner (HP) and quantified using Fiji (ImageJ 1.53t) software.

### 2.8. H_2_S and Persulfidation Measurement

Free H_2_S was measured in cells using the SF_7_-AM fluorescent probe [30] (Sigma-Aldrich). The probe was dissolved in anhydrous DMF at 5 mM and used at 5 μM in serum-free RPMI. Live-cell image acquisition was performed using a Nikon Ti2 spinning disk confocal microscope. Global protein persulfidation was assessed on VSMC grown on glass coverslips as previously described [9]. Cells were incubated for 20 min with 1 mM 4-Chloro-7-nitrobenzofurazan (NBF-Cl, Sigma-Aldrich) diluted in PBS. Then, cells were washed with PBS and fixed for 10 min in ice-cold methanol. Coverslips were rehydrated in PBS and incubated with 1mM NBF-Cl for 1 h at 37 °C. In parallel, a Daz2-Cy5.5 solution was prepared by mixing 1mM Daz-2, 1 mM alkyne Cy5.5, 2 mM copper(II)-TBTA, 4mM ascorbic acid and incubating overnight at RT, followed by quenching for 1h with 20mM EDTA. Fixed cells were further incubated at 37 °C for 1h in the Daz2-Cy5.5 solution. Finally, coverslips were washed 3 times in methanol and 2 times in PBS, mounted in Vectashield mounting medium with DAPI, and visualized with a 90i Nikon fluorescence microscope. Persulfidation was measured as the ratio of Daz2-Cy5.5 over NBF-Cl signal per cell by two independent experimenter blinded to the conditions using the Fiji (ImageJ 1.53t) software.

### 2.9. BrdU Assay

VSMCs were grown at 80% confluence (5 × 10^3^ cells per well) on glass coverslips in a 24-well plate and starved overnight in serum-free medium. Then, VSMC were either treated or not (ctrl) with the drug of choice for 24 h in full medium (RPMI 10% FBS) in the presence of 10 µM BrdU (Sigma-Aldrich). All conditions were tested in parallel. All cells were fixed in ice-cold methanol 100% after 24 h of incubation and immunostained for BrdU. Images were acquired using a Nikon Eclipse 90i microscope (Nikon Europe B.V., Egg/ZH, Switzerland). BrdU-positive nuclei and total DAPI-positive nuclei were automatically detected using the Fiji (ImageJ 1.53t) software [14].

### 2.10. Wound Healing Assay

VSMCs were grown at confluence (10^4^ cells per well) in a 12-well plate and starved overnight in serum-free medium. Then, a scratch wound was created using a sterile p200 pipette tip and medium was changed to full medium (RPMI 10% FBS) in presence of 0.5 µg/mL mitomycin C (Sigma-Aldrich; Merck KGaA, Darmstadt, Germany) to block proliferation. Repopulation of the wounded areas was recorded by phase-contrast microscopy in a Nikon Ti2-E live cell microscope (Nikon Europe B.V., Egg/ZH, Switzerland). All conditions were tested in parallel. The area of the denuded area was measured automatically using the macro Wound_healing_size_tool_updated.ijm [31] in the Fiji (ImageJ 1.53t) software. Data were expressed as a percentage of the wound closure. Morphometric measurement of cell geometry was performed manually by two independent experimenters blinded to the conditions using the shape descriptors in measurements in Fiji (ImageJ 1.53t) software.

### 2.11. Statistical Analyses

All experiments adhered to the ARRIVE guidelines and followed strict randomization. All experiments and data analysis were conducted in a blind manner using coded tags rather than the actual group name. All experiments were analyzed using GraphPad Prism 9. Normal distribution of the data was assessed using Kolmogorov–Smirnov tests. All data with normal distribution were analyzed by unpaired bilateral Student’s *t*-tests or Mixed-effects model (REML) followed by post-hoc *t*-tests with the appropriate correction for multiple comparisons when comparing more than 2 groups. For non-normal distributed data, Kruskal–Wallis non-parametric ranking tests were performed, followed by Dunn’s multiple comparisons test to calculate adjusted *p* values. Unless otherwise specified, *p*-values are reported according to the APA 7th edition and New England Journal of Medicine statistical guidelines. * *p* < 0.033, ** *p* < 0.002, *** *p* < 0.001.

### 2.12. Ethics Statement

Human vein and artery segments were obtained from donors who underwent vascular surgery at the Lausanne University Hospital. Written informed consent was obtained from all donors. The study protocols for organ collection and use were reviewed and approved by the Lausanne University Hospital (CHUV) and the Cantonal Human Research Ethics Committee (CER-VD, no IRB number, Protocol Number 170/02), and are in accordance with the principles outlined in the Declaration of Helsinki of 1975, as revised in 1983 for the use of human tissues.

## 3. Results

### 3.1. CSE Is Expressed in the Media and Intima of Human Saphenous Vein and Artery Segments

CSE protein was assessed in segments of healthy human vein and artery. Artery segments expressed higher levels of CSE, eNOS, CBS and 3MST than vein segments as measured by WB analysis (Figure 1a). However, this did not translate in higher H_2_S production capacity as assessed by led acetate assay (Figure 1b).

Further CSE immunohistology shows that CSE was expressed both in the media and intima layers of the artery segments (Figure 2a,b). CSE was also expressed in the endothelium of the small vessels of the vasa vasorum of arteries, easily identified using a Von Willebrand factor (VWF) immunostaining of EC (Figure 2a,b). In human veins, CSE was mostly detectable in the media layer, but not so much in the endothelium (Figure 2c). CSE was also expressed in the endothelium of the small vessels of the vasa vasorum (Figure 2c). In veins, CBS was expressed mostly in the media layer (Figure 2c). In artery segments, 3-MST was detectable in the media, but was mostly expressed by the EC of the intima and smaller vessels of the vasa vasorum. Interestingly, some arteries featured benign intimal hyperplasia and 3-MST was not expressed in this neointima layer (Appendix A). In contrast, CSE was similarly expressed throughout the media and neointima layers. In the saphenous vein, 3-MST was not expressed in the media layer, mainly detected in the vasa vasorum vessels of the media and adventitia layers, and seldom detected in the EC from the intima (Appendix A).

### 3.2. CSE Expression Is Regulated by Flow

To study the regulation of CSE expression in response to flow, we next investigated CSE expression in a model of static human vein culture. Static vein culture leads to formation of IH as previously described [9,13,14], as assessed by VGEL staining (Figure 3a). qPCR analysis revealed that CSE mRNA expression was increased 4-fold after 7 days in static culture (Figure 3b). CSE overexpression correlated with heme oxygenase 1 (HO-1) and thioredoxin 1 (TRX1) overexpression, which are known target of H_2_S [32,33,34] (Figure 3b). CSE overexpression was also accompanied by overexpression of Activating Transcription Factor 4 (ATF4), a transcription factor known to stimulate CSE transcription [14,29], and Kruppel-like factor 2 (KLF2) down-regulation (Figure 3b). CBS mRNA expression was also increased about 4-fold, whereas MPST mRNA expression increased about 2-fold (Figure 3b). WB analysis confirmed CSE 4-fold protein overexpression in static culture, whereas CBS protein levels were decreased by about 30%, and 3-MST was increased about 1.5-fold (Figure 3c). The increase in CSE expression was accompanied by an increase in H_2_S production, as measured by lead acetate (Appendix A). IHC analyses confirmed that CSE expression was increased in the media layer of human veins after static culture (Figure 3d). Of note, CSE expression seemed reduced in the endothelium after 7 days in static culture (Figure 3d). CBS expression was reduced and restricted to fewer cells expressing higher CBS levels in the media and CBS expression was not detectable in the endothelium after 7 days in static culture (Figure 3d). 3-MST expression was lost in the EC of the intima, still detectable in the EC of the vasa vasorum, as well as in the media layer (Appendix A).

To selectively evaluate the role of pressure and shear stress on CSE expression, we further assessed CSE, 3-MST and CBS expression in human saphenous veins segments perfused under pulsatile low pressure (LP; venous regimen mean pressure = 7 mmHg) or high pressure (HP; arterial regimen mean pressure = 100 mHg). Arterial (HP), but not LP, perfusion stimulated thinning of the media layer and the formation of IH, as assessed by VGEL and Herovici staining (Figure 4a), as previously described [26]. After 7 days in LP conditions, CTH and CBS mRNA expression remained unchanged, whereas MPST and eNOS mRNA expression were reduced 4-fold (Figure 4b). After 7 days in HP condition, MPST and eNOS were further reduced and CTH expression was reduced 3-fold, whereas CBS mRNA expression increased 2-fold (Figure 4c). There was a strong correlation between MPST and eNOS mRNA expression across conditions (Figure 4d), whereas there is no correlation between CTH and eNOS mRNA expression (Figure 4e). We further looked at CSE and CBS protein expression in human vein segments perfused at high pressure (HP). Western blot analysis revealed that both CSE and CBS protein levels were reduced in those conditions compared to the native vein (Figure 4f). IHC staining confirmed that CSE expression was decreased by HP throughout the media and intima layers (Figure 4g). Similar to what we observed in static vein culture (Figure 3d), the CBS protein expression was reduced and restricted to fewer cells expressing higher CBS levels in the media. CBS was not detectable in the endothelium (Figure 4g). In contrast with static vein culture, 3-MST expression remained undetectable in the media layer. 3-MST was still detectable in the EC of the small vessels of the vasa vasorum (Appendix A).

### 3.3. CSE Regulates Human VSMC Migration

CSE expression in the media of large human vein was decreased under pulsatile arterial high pressure (Figure 3 and Figure 4). We then tested the role of CSE in VSMC proliferation and migration in primary human VSMC derived from great saphenous vein segments. Adenoviral-mediated *CSE* overexpression increased CSE protein expression in primary VSMC but did no impact CBS and 3MST expression (Figure 5a). Adenoviral-mediated GFP expression indicated that the infection efficiency of VSMC was above 80% (Appendix A). As expected, *CSE* overexpression increased H_2_S production (SF_7AM_; Figure 5b) and protein persulfidation (Figure 5c). In contrast, siRNA-mediated *CSE* silencing reduced CSE protein level by 60–70% (siCTH^1^ = 0.37 ± 0.09, *p* = 0.008; siCTH^2^ = 0.45 ± 0.05, *p* = 0.02 by mixed effect model with Šídák’s multiple comparisons test), without impacting CBS or 3MST protein levels (Figure 5d). Transfection efficiency in VSMC was above 90% as assessed using siGLO transfection (Appendix A).

*CSE* overexpression or knock-down had no impact on VSMC proliferation as assessed by BrdU incorporation (Figure 6a,b), and apoptosis (Figure 6c,d). In contrast, *CSE* overexpression slowed down migration in a wound healing assay (Figure 6e), whereas *CSE* knock-down accelerated wound healing (Figure 6f). Of note, *CSE* overexpression also increased the circularity and reduced the feret diameter of the VSMC, whereas *CSE* knock-down reduced circularity and increased the feret diameter, indicating that CSE modulation had an impact on the geometry of the cells associated with migration (Figure 6g,h). *CSE* overexpression caused the most prominent changes, characterized by reduced spindle shape morphology due to a more static phenotype. In contrast, CSE knocked-down cells were more mobile and more elongated (decreased circularity and increased feret diameter).

CSE and H_2_S have been proposed to regulate cell proliferation and migration via the ERK1,2 kinase and the mTOR pathways in VSMC [35]. Here, CSE overexpression or CSE knock-down had no effect on P-ERK in VSMC in our experimental settings. CSE overexpression or CSE knock-down also had no effect on the phosphorylation of the S6 ribosomal protein (S6RP), a downstream target of the PI3K/Akt/mTOR pathway (Appendix A).

## 4. Discussion

Endogenous H_2_S production in mammals results from the oxidation of the sulfur-containing amino acids cysteine and homocysteine via the reverse “trans-sulfuration” pathway mainly via CSE and CBS, and 3-MST. Although the enzymes and pathways responsible for H_2_S production are well described, the regulation of these genes in human vascular diseases remains largely unknown. In this study, we investigated the expression of CSE, CBS and 3-MST in segments of human saphenous vein and artery. CBS was mainly detected in the media layer of human vessels. In contrast, 3-MST was mainly detected in the endothelium and in the EC of small vessels of the vasa vasorum in both arteries and veins. This suggests a more prominent role of 3-MST in EC of small caliber vessels and capillaries, which is consistent with previous studies suggesting a key role for 3-MST in EC [19]. In line with this hypothesis, 3-MST expression closely correlates with eNOS expression in our model of ex vivo culture. 3-MST is also expressed in the media of arteries, but not the media of veins, indicating differential regulation of 3-MST in arterial and venous vessels. Rodent studies suggest that CSE is mainly expressed in EC in the cardiovascular system and that the endothelium is the main source of H_2_S in blood vessels [15,16,17,36]. That said, CSE has also been described in VSMC and proposed to be a functional, albeit minor source of H_2_S [8,18,37]. Here, CSE was expressed in the endothelium of large vessels, and in small vessels of the vasa vasorum in native human artery and vein. CSE was also abundant in the media layer.

We then investigated the regulation of the three enzymes in vein segments placed in ex vivo culture. CBS protein expression was reduced in vein segments placed in pathological culture conditions ex vivo, both in the absence of flow and in high pressure flow. This indicates that shear stress does not regulate CBS expression in VSMC. Interestingly, although CBS protein levels were lower, CBS mRNA levels were increased in ex vivo culture, suggesting a differential regulation of mRNA and protein expression and a possible effect of ex vivo culture on protein stability. However, histological analysis suggested a more complex regulation, as CBS was no longer detected in most cells, whereas it was overexpressed in a few cells present in the media and neointimal layer. Further studies are required to determine which cells overexpress CBS when most cells in the media and neointima appear to down-regulate CBS expression. This is of particular interest as adult VSMC are highly plastic cells [38], and the switch from a quiescent ‘contractile’ phenotype to a proliferative ‘synthetic’ phenotype plays a major role in the context of IH [39]. Recent VSMC lineage tracing studies in mice using in vivo cell fate tracing with SMC-specific genetic reporter tools suggest that a small subset of VSMCs expand after injury to form clonal patches of neointimal cells [40,41,42]. Further studies are required to elucidate the identity of this small subset of VSMCs in human tissue, and whether CBS is expressed or not in this subset in the context of IH.

3-MST mRNA and protein expression were severely reduced by ex vivo perfusion of vein segments, independent of flow and shear stress. Interestingly, 3-MST expression was mostly restricted to EC in veins and closely correlated with eNOS expression in our model of ex vivo culture, which make sense as our model results in endothelial dysfunction and rapid loss of endothelial-specific markers [25,26,43]. However, static ex vivo culture tended to stimulate 3-MST mRNA and protein expression despite severe endothelial dysfunction. This suggests that 3-MST may be negatively regulated by shear stress in a similar way to CSE. 3-MST was largely undetectable in the media layer of veins, but it was detectable in the media of arteries, suggesting that 3-MST plays a role in arterial VSMC, but that high shear stress per se does not negatively regulate 3-MST expression. 3-MST was overexpressed in VSMC in veins in static condition and in cultured VSMC in vitro, suggesting that 3-MST could be involved in VSMC reprogramming in the context of IH. However, arterial perfusion prevented 3-MST expression, so it is unlikely that 3-MST play a main role in IH in vivo. In a recent study, 3-MST was found to be expressed in VSMC and cardiomyocytes, and 3-Mst^−/−^ mice were protected against myocardial ischemia-reperfusion injury [44]. Further studies are required to better characterize the role and regulation of 3-MST in VSMC.

About CSE, our data confirm that CSE is negatively regulated by shear stress. Using our model of ex vivo vein perfusion, we observed that high pressure inhibits CSE expression both in the media and in the endothelium. In contrast, static ex vivo vein culture stimulates CSE expression and H_2_S production. In those conditions, CSE overexpression also correlated with HO-1 and TRX1 overexpression. HO-1 is a direct target of nuclear factor (erythroid-derived 2)-like 2 (NRF2), and H_2_S promotes the NRF2 anti-oxidant response via persulfidation of Kelch-like ECH-associated protein 1 (Keap1), which leads to NRF-2-induced expression of several proteins including HO-1 [32]. TRX1 expression is also stimulated by H_2_S [33,34]. These data infer that there could be a functional increase in CSE expression in vein culture under static conditions. However, HO-1 or TRX1 expression are sensitive to oxidative stress and likely increased in response to the ex vivo environment. Further studies are needed to assess redox status and oxidative stress in our model. Mechanistic studies are also needed to evaluate the contribution of H_2_S to HO-1 or TRX1 expression in those conditions.

The fact that static culture stimulates CTH expression is in line with previous evidence showing that CSE expression is downregulated by high shear stress and is predominantly found in regions of disturbed flow [20,21]. However, we did not observe higher levels of CSE in native human vein segments compared to aortic segments, despite the high shear stress in arteries. In fact, CSE, CBS and 3MST levels were higher in arteries than in veins. However, this could be due to a higher cell content in arterial tissue compared to venous tissue, which contains more connective tissue than arteries. Consistent with this hypothesis, H_2_S production as measured by the lead acetate assay was similar in veins and arteries. Interestingly, this regulation by shear stress is not unique to EC and was also observed in VSMC. The close correlation between CSE mRNA and protein levels suggests a regulation at the transcriptional level. CTH expression can be induced by stress factor such as endoplasmic reticulum stress, amino acid restriction or oxidative stress via ATF4 [14,29]. Here, ATF4 was overexpressed in response to static vein culture, which may drive CSE overexpression in that condition. The micro RNA miR-27b, which is highly expressed in EC and involved in angiogenesis [45,46], has been described to down-regulate CTH expression [20]. However, this mechanism has been described only in EC and miR-27b is down-regulated in EC in pathological condition [20]. miR-27b is under the control of KLF2, a mechanosensitive transcription factor induced by laminar shear stress participating in vascular development [20]. Here, KLF2 expression was severely repressed in static culture, which may result in loss of miR-27b and CTH overexpression. That said, there is no report of miR-27b in VSMC or in the context of IH and further studies are needed to evaluate whether miR-27b in VSMC inhibit CTH expression in relation with flow.

Overall, we observed that all the three main enzymes involved in H_2_S production are expressed in the vessels but with different patterns of expression and regulation between vein and arteries and between VSMC and EC. We also report that CTH and CBS are largely expressed in the media layer although most of the H_2_S research has been focused on EC. The pattern of expression of CBS in the media upon ex vivo culture is of particular interest in the context of IH and warrants further investigation.

Given that CSE is highly expressed in VSMC and downregulated by high-pressure perfusion in veins, we further investigated the role of CSE in human primary venous VSMC. We document that CSE is a specific modulator of VSMC function independent of EC or EC-derived H_2_S production, and that CSE controls the migration of primary human VSMC. This agrees with studies showing that VSMCs isolated from Cse^−/−^ mice are more motile than their WT counterparts, and that blocking Cse activity with PAG in VSMC increases cell migration [8,18]. We and others reported that H_2_S donors inhibits VSMC proliferation [9,12,13,14,35]. Here, we did not observe a significant effect of CSE on cell proliferation. This contrasts with a previous study showing that Cse overexpression decreased proliferation and even induced VSMC apoptosis. This discrepancy probably results from the level of Cse overexpression, which was probably higher in the previous study by Yang et al. [35]. This study employed arterial VSMC, whereas we utilized venous VSMC. Venous and arterial VSMC exhibit distinct properties; therefore, the origin of VSMC (venous versus arterial) could have contributed to the differences observed in our results. It is also likely that increased CSE expression and activity, which leads to greater H_2_S production, induces cell apoptosis since high levels of H_2_S are known to cause cell cycle arrest and apoptosis. [47]. Here, we did not observe any toxic effect of CTH knockdown or overexpression on VSMC, supporting the refinement of our experimental design.

The mechanisms whereby H_2_S affect VSMC proliferation and/or migration are not fully understood. Cytotoxic CSE overexpression or exogenous H_2_S supplementation induces VSMC cycle arrest and apoptosis by stimulating ERK1/2, p38 MAPK and p21 Cip [35]. Exogenous H_2_S donor treatment in VSMC has also been shown to inhibit the MAPK pathway, especially ERK1,2, and the mTOR pathways, which correlates with reduced VSMC proliferation and migration [13]. Here, CSE manipulation has no effect on the phosphorylation of ERK and S6RP, a downstream target of mTOR. This probably reflects the more subtle CSE variation in our experimental design compared with exogenous H_2_S supply or high Cse overexpression in previous studies. Rather than a physiological response to CSE-derived H_2_S, the effect of CSE overexpression on VSMC proliferation may be due to cell cycle arrest associated with a cytotoxic effect of H_2_S. In accordance with this hypothesis, it was previously shown that inhibition of ERK did not prevent the effect of NaHS on VSMC migration [8], whereas ERK inhibition was instrumental in the effect of Cse and H_2_S on VSMC proliferation [35]. Of note, several studies in EC also reported that Cse regulates cell migration, but not proliferation [14,17]. We propose that low levels of H_2_S affect VSMC migration without affecting their proliferation. Our main finding is that CSE variations reshape VSMC, suggesting an effect on the cytoskeleton and interaction with the ECM. This is consistent with our previous finding that H_2_S donors inhibit microtubule polymerization in VSMC [9], and the findings that CSE deficiency in mouse VSMC results in increased expressions of β1-integrin and increased migration [8]. In EC, it was recently shown that integrins are extensively sulfhydrated, and that β3 integrin S-sulfhydration promotes adhesion and is required for EC alignment with flow [48]. In this study, they further demonstrated that Cse deficiency in EC leads to overactivation of RhoA, a major hub regulating cell migration and adhesion. Interestingly, β3-integrin is also expressed in VMSC and β3-integrin signaling is instrumental for enhanced VSMC proliferation and migration in vascular disease [49,50]. We hypothesize that CSE-mediated integrin sulfhydration promotes VSMC adhesion, thereby limiting migration. Further studies are needed to determine the exact role and balance between β1- and β3-integrins and RhoA signaling in the regulation of cell migration by CSE in VSMC.

### 4.1. Limitations

This study has some limitations. First, the study used discarded anonymous arterial and venous samples. Although the regulation of CSE was robust under high shear stress, analysis of samples collected from a specific study could provide more insight into how individual genetic or environmental factors affect the expression of CSE and other enzymes. This method would allow correlation analysis and a better understanding of the effects of pre-existing conditions, age and sex. Second, we could not collect any failed arteriovenous bypasses to confirm the H_2_S enzyme expression in vivo compared to the ex vivo arterial perfusion setup. Finally, we hypothesized that H_2_S release by CSE inhibits VSMC migration by persulfidating cysteine residues in tubulin proteins, leading to microtubule depolymerization as described in earlier studies [9]. We also suggest that integrin persulfidation, which is promoted by CSE, stimulates VSMC adhesion and restricts migration. Further research is needed to test these hypotheses and to demonstrate persulfidation of cysteine residues in target proteins such as tubulin and integrins. Furthermore, H_2_S may modify other proteins involved in cytoskeleton dynamics, which can also contribute to the effect of CSE on VSMC migration. Additionally, CSE may act independently of H_2_S and persulfidation, through an unknown mechanism. Recently, it was proposed that CSE works as a scaffold protein preventing p53 translocation to the nucleus in the context of EC aging, independently of H_2_S production [51].

### 4.2. Conclusions

Bypass IH after cardiovascular surgeries is a significant complication. Currently available therapies to reduce IH are limited. They also negatively affect endothelial recovery, reducing their long-term efficacy and prolonging the need for antithrombotic therapy. Novel strategies to inhibit VSMC proliferation while promoting EC recovery are needed. In this context, the gasotransmitter H_2_S is a promising candidate as CSE/H_2_S inhibit VSMC migration and IH, while stimulating EC migration and promoting endothelium repair [3,52]. Designing chemical compounds that allow for controlled release of H_2_S is difficult due to its instability and short half-life. Currently, there is no clinically approved H_2_S-releasing molecule. A better understanding of endogenous H_2_S production could allow us to design new strategies to improve endogenous H_2_S production. Here, we demonstrate that all three main enzymes involved in H_2_S production are expressed in vessels, albeit with different patterns of expression and regulation. Through the use of an ex vivo vein perfusion system that mimics venous and arterial pressure and flow, we determined that high-pressure perfusion decreases CSE expression in segments of the human saphenous vein. We also confirmed that CSE specifically regulates VSMC migration, probably via regulating the cytoskeleton. Our data also call for further investigation of the role CBS in VSMC specifically in the context of IH.

In summary, our experiments revealed that high shear stress in arteriovenous bypass grafts inhibits CSE expression. This suggests that CSE down-regulation occurs in vivo in bypass grafts, thereby contributing to VSMC migration and graft IH.

## Figures and Tables

**Figure 1 antioxidants-12-01731-f001:**
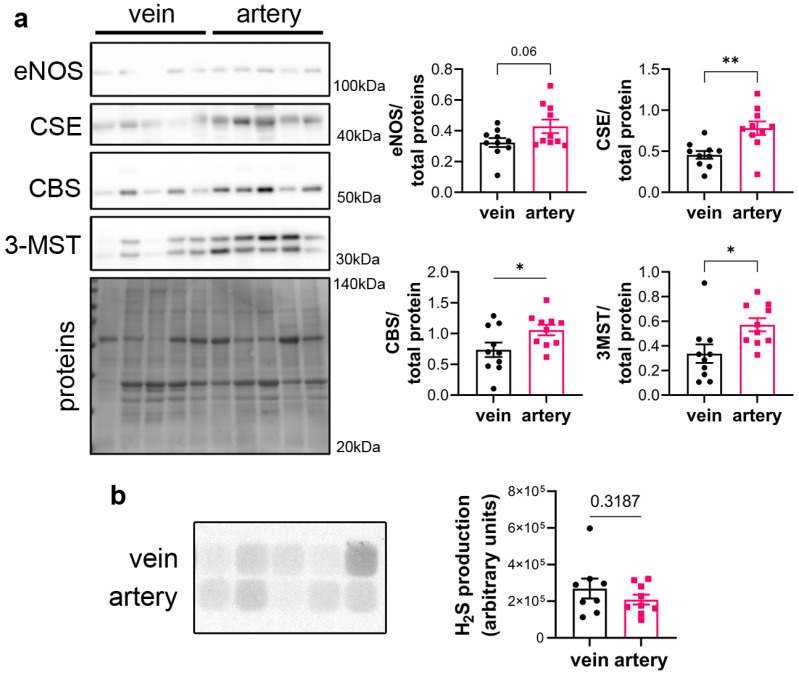
**CSE is expressed in human artery and saphenous vein segments.** (**a**) Representative Western blot and quantitative assessment of eNOS, CSE, CBS and 3-MST from freshly isolated human arteries and saphenous veins. Data are mean ± SEM of 10 arteries and 9 veins. (**b**) Lead acetate assay in freshly isolated human arteries and saphenous veins. Data are mean ± SEM of 9 arteries and 8 veins. * *p* < 0.033, ** *p* < 0.002, as determined by bilateral paired *t*-test.

**Figure 2 antioxidants-12-01731-f002:**
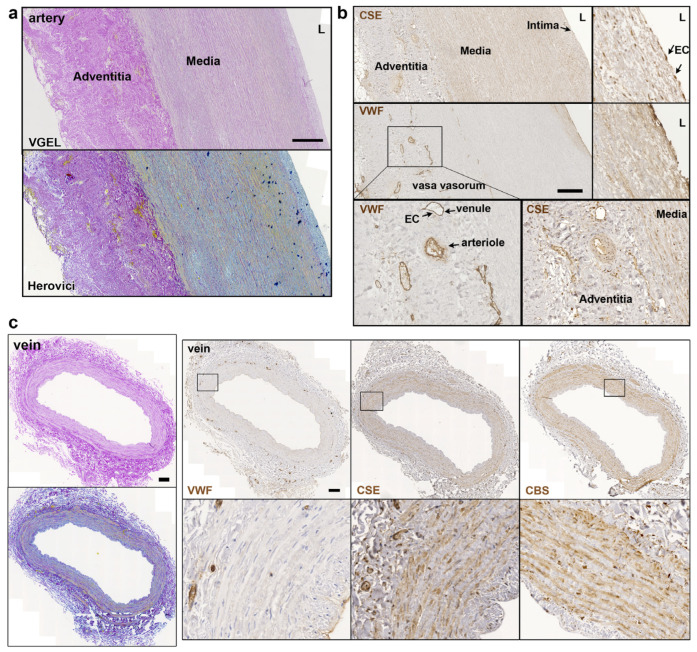
**CSE is expressed in the media and intima of human artery and saphenous vein.** Representative, VGEL, Herovici, CSE, CBS and VWF staining as indicated in section of a human artery (**a**,**b**) and a saphenous vein. Insets are 2.5 fold magnification (**c**). Images are representative of 8 arteries and 10 saphenous veins. Insets from black boxes are 5 fold magnification Scale bar = 100 µm. L = lumen.

**Figure 3 antioxidants-12-01731-f003:**
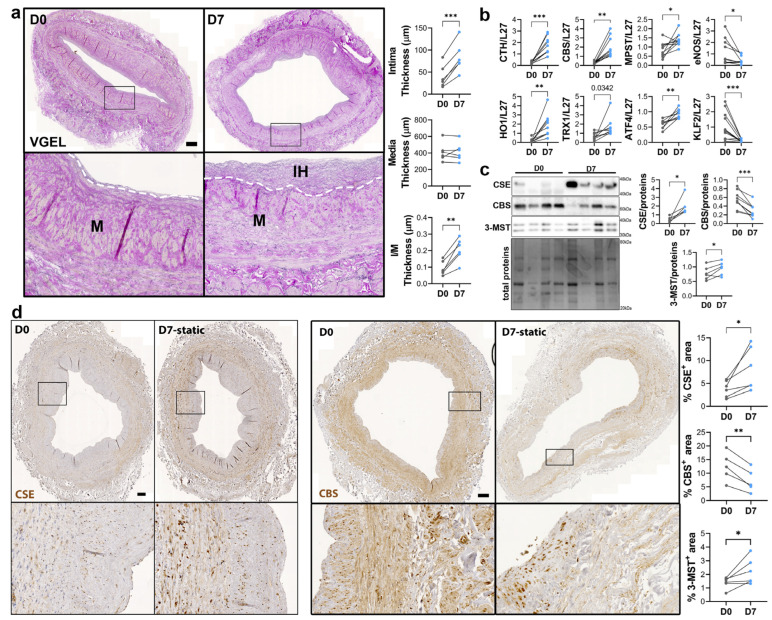
**Static culture significantly increases CSE expression in human vein segments.** (**a**) Representative histological VGEL staining and morphometric measurements of intima thickness, media thickness and intima over media ratio of freshly isolated human vein segments (D0) or after 7 days (D7) in static culture. Scale bar 100 µm. The white dotted lines highlight the separation between the media and neointima regions. Black boxes outline the regions for 4-fold insets magnification (**b**–**d**) normalized mRNA (**b**) and protein expression as assessed by Western blotting (**c**) or immunohistochemistry (**d**) in freshly isolated human vein segments (D0) or after 7 days (D7) in static culture. Scale bar 100 µm. Black boxes outline the regions for 4-fold insets magnification. 5 to 8 different veins/patients. * *p* < 0.033, ** *p* < 0.002, *** *p* < 0.001, as determined by bilateral paired *t*-test.

**Figure 4 antioxidants-12-01731-f004:**
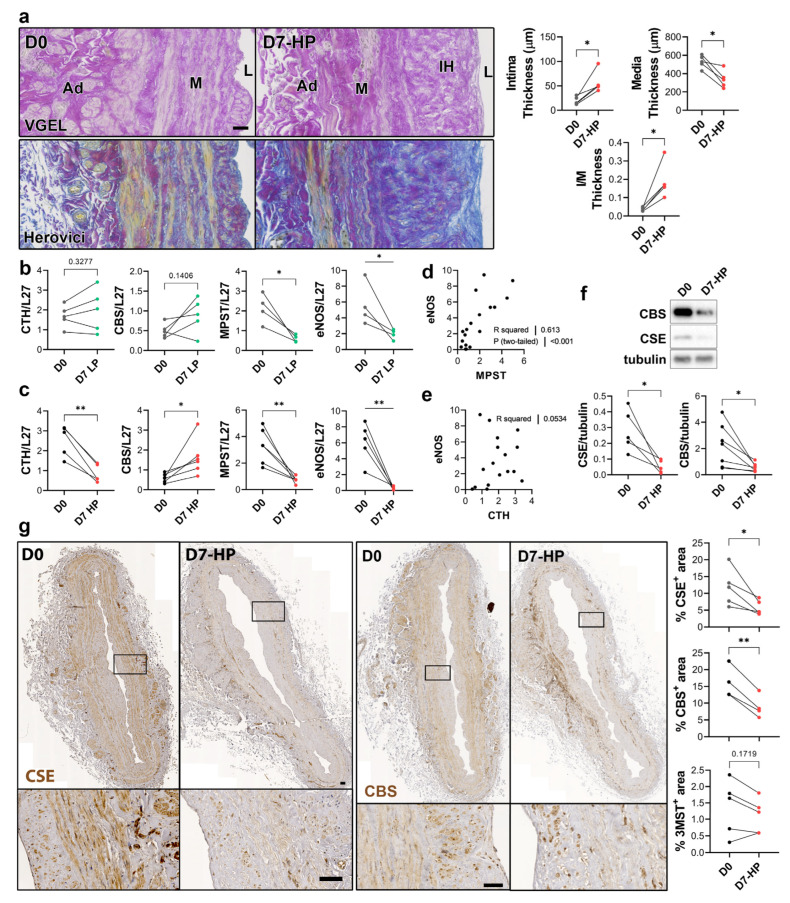
**High pressure perfusion decreases CSE expression in human vein segments.** (**a**) Representative histological sections (left panels) stained for elastin (VGEL) and collagen (Herovici) and morphometric measurements (right panels) of intima thickness, media thickness and intima over media ratio of freshly isolated human vein segments (D0) or in veins exposed to pulsatile high pressure (D7-HP; mean = 100 mmHg) perfusion for 7 days. Scale bar 50 µm. (**b**,**c**) Normalized CSE, CBS and 3-MST (MPST), eNOS mRNA (**b**,**c**). (**d**,**e**) Pearson’s correlation coefficient between eNOS and MPST (**d**) or CTH (**e**) mRNA levels. (**f**,**g**) Protein expression as assessed by Western blotting (**d**) or immunohistochemistry (**g**) in freshly isolated human vein segments (D0) or in veins exposed to pulsatile low pressure (D7-LP; mean = 7 mmHg) or high pressure (D7-HP; mean = 100 mmHg) perfusion for 7 days. Insets are 5-fold magnification from black boxes. Scale bar 100 µm. 4 to 5 different veins/patients. * *p* < 0.033, ** *p* < 0.002, as determined by bilateral paired *t*-test.

**Figure 5 antioxidants-12-01731-f005:**
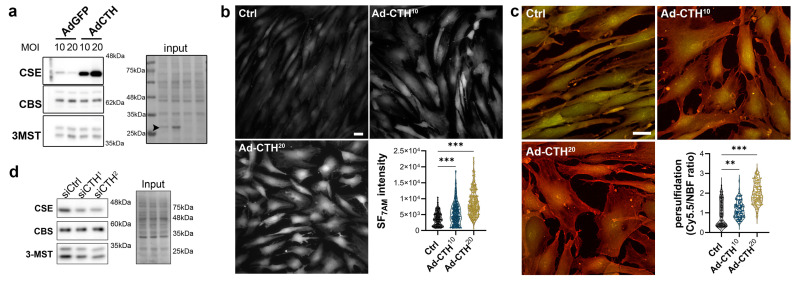
***CSE* overexpression in human VSMC increases H_2_S production and protein persulfidation.** (**a**) CSE, CBS and 3-MST protein expression 48 h post VSMC infection with an adenovirus encoding GFP (AdGFP) or CSE (AdCTH) at MOI 10 or 20, as indicated. Arrowheads in input indicate GFP expression upon Ad-GFP infection. Data are representative of 6 independent experiments. (**b**) Live-cell imaging of H_2_S production using the SF_7AM_ probe in VSMC infected, or not (Ctrl), with an adenovirus encoding *CSE* at MOI 10 or 20, as indicated. Images are representative of 5 independent experiments. Quantitative assessment (violin plots) of SF_7AM_ fluorescent in individual cells across 5 experiments. (**c**) In situ labelling of intracellular protein persulfidation assessed by DAz-2: Cy5.5 (red), normalized to NBF-adducts fluorescence (green), in VSMC infected, or not (Ctrl), with an adenovirus encoding *CSE* at MOI 10 or 20, as indicated. Data are representative of 5 independent experiments. Representative images of 5 independent experiments. Violin plots of DAz-2: Cy5.5 over NBF fluorescence in individual cells across 5 experiments. (**b**,**c**) Scale bar 20 μm. ** *p* < 0.002, *** *p* < 0.001 report adjusted *p*-values as determined by Kruskal–Wallis non-parametric ranking followed by Dunn’s multiple comparisons tests. (**d**) Western blot analyses of CSE, CBS and 3-MST expression 48 h post VSMC transfection with a control siRNA (siCtrl) or two distinct CTH siRNA (siCTH^1^ and ^2^), as indicated. Data are representative of 6 independent experiments.

**Figure 6 antioxidants-12-01731-f006:**
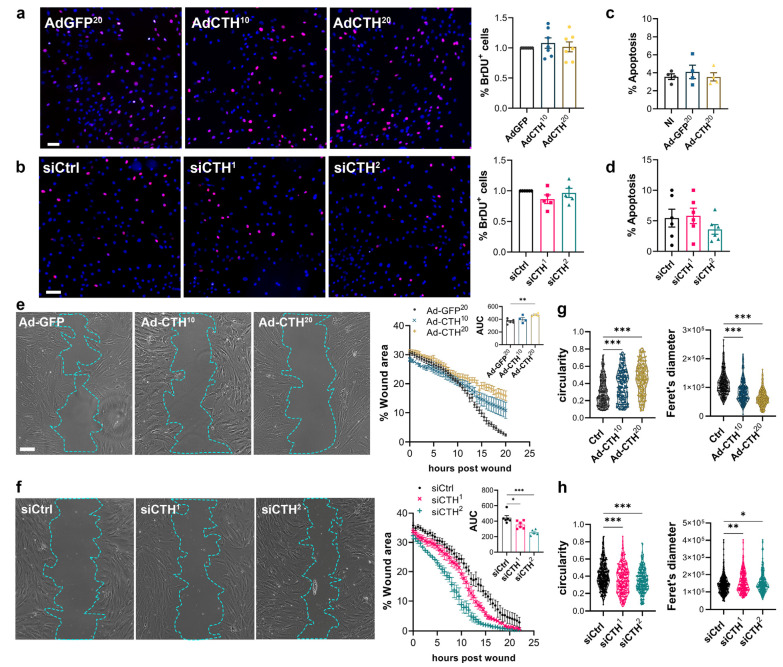
**CSE inhibits VSMC migration, but not proliferation in vitro.** (**a**) VSMC proliferation (BrdU incorporation) in VSMC infected with an adenovirus encoding GFP (AdGFP) or CSE (AdCTH) at MOI 10 or 20, as indicated. (**b**) VSMC proliferation (BrdU incorporation) in VSMC transfected with a control siRNA (siCtrl) or two distinct CTH siRNA (siCTH^1^ and ^2)^, as indicated. (**a**,**b**) Data are % of BrdU positive nuclei (pink) over DAPI positive nuclei. Scale bar: 50 µm. Data shown as mean ± SEM of 5 to 6 independent experiments. (**c**,**d**) % apoptosis in infected VSMC (**c**) or in transfected VSMC as indicated (**d**). Data shown as mean ± SEM of 4 to 5 independent experiments. (**a**–**d**) No statistical differences as determined by repeated measures one-way ANOVA with Dunnett’s multiple comparisons tests. (**e**) VSMC migration (wound healing) in VSMC infected with an adenovirus encoding GFP (AdGFP) or CSE (AdCTH) at MOI 10 or 20, as indicated. (**f**) VSMC migration (wound healing) in VSMC transfected with a control siRNA (siCtrl) or two distinct CTH siRNA (siCTH^1^ and ^2^), as indicated. (**e**,**f**) Left panels: Bright field images of VSMC 12 h post wound. Scale bar: 50 µm. Right panels: Data are mean ± SEM of the percentage of wound closure in 5 to 6 independent experiments. Insets show area under the curve (AUC) of wound healing. * *p* < 0.033, ** *p* < 0.002, *** *p* < 0.001, report adjusted *p*-values as determined by one-way ANOVA followed by Dunnett’s multiple comparisons tests. (**g**,**h**) Cell morphology during wound healing is expressed as the circularity index and the Feret’s diameter shown as violin plots of individual cells across 5 independent experiments. * *p* < 0.033, ** *p* < 0.002, *** *p* < 0.001 report adjusted *p*-values as determined by Kruskal–Wallis non-parametric ranking followed by Dunn’s multiple comparisons tests.

## Data Availability

The data presented in this study are available on request from the corresponding author.

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
