# Peer review of "Cystathionine Gamma Lyase Is Regulated by Flow and Controls Smooth Muscle Migration in Human Saphenous Vein"

_antioxidants, 2023, doi:10.3390/antiox12091731_

Round 1

Reviewer 1 Report

H2S signaling was described previously to contribute in the regulation of endothelial functions. However, this manuscript provides new evidence for the role of H2S biology in cardiovascular system that is based on decent experimental design covering human-samples based methodological approach. Thus, the concept is scientifically interesting. Manuscript is also technically well-prepared. Nevertheless, I would like to point out few concerns listed below.

Major Comments:

Authors showed the alterations of the enzymes expressions/H2S levels under variable pressure – this is an asset. Nevertheless, I have not noticed any data that would directly refer to the oxidation or redox signaling. Following this issue but also taking into account the profile of the journal, some representative data (e.g. on the impact of altered CSE, CBS or MPST expression on the oxidative endothelial damage would be scientifically valuable (at least in one experimental condition covered by the experimental design)).

Authors discussed (and mentioned in the Results section) mTOR activity but there is no direct evidence in this manuscript (this particular experimental approach) on the correlation between phosphorylation/activity of mTOR pathway in particular segments or experimental groups and/or CSE, CBS, MPST down- or upregulation. Evaluation of mTOR phosphorylation (not only limited to e.g. ERK) is recommended or at least, revision of the conclusions and the way of the story telling is required.

I recommend to expand the conclusions and to explain more thoroughly in this part what exactly Authors discovered here and what is the new functional/fundamental approach on H2S signaling in cardiovascular system based on their data. Other words, Authors should describe the general outcome in more details and incorporate it to current cardiovascular physiology. I think that this could help to expose more efficiently the scientific value of this research to the readers.

Minor comments:

Lead acetate (CSE activity assay): please, add more technical details and the reference to previous publications if applicable. It seems that this assay is not CSE-specific – it does not exclude the impact of CBS activity on the final measurement.

Discussion: “Although the enzymes and pathways responsible for H2S production are well described, little is known about their regulation in pathophysiologic conditions” – please, revise the sentence and this part of the Discussion in general; there are plethora of publications showing altered activity of all 3 H2S-producing enzymes under pathological conditions.

Reviewer 2 Report

Cystathionine gamma-lyase (CSE) is an enzyme that produces hydrogen sulfide (H2S), a gasotransmitter that inhibits intimal hyperplasia (IH), a vascular complication of bypass surgery. The authors investigate the expression and regulation of CSE and other H2S-producing enzymes in human saphenous vein and artery segments. The authors show that CSE is expressed in the media, neointima, and intima of the vessels, and is negatively regulated by arterial shear stress. CSE inhibits primary human vascular smooth muscle cell (VSMC) migration but not proliferation. CBS and 3-MST are also expressed in the vessels but with different patterns and regulations. The authors use ex vivo vein perfusion systems to mimic venous or arterial pressure and flow. They find that static culture or low-pressure perfusion increases CSE expression, while high-pressure perfusion decreases CSE expression in vein segments. They propose that high shear stress in arteriovenous bypass grafts inhibits CSE expression, possibly contributing to increased VSMC migration and IH formation. As an academic reviewer, I would like to provide the following questions or comments:

1.          The authors have provided a comprehensive and novel study on the expression and regulation of H2S-producing enzymes in human vessels and their role in IH. The study is well-designed, executed, and presented, and the results are of high interest and relevance to vascular biology and surgery.

2.          The study protocols for organ collection and use were reviewed and approved by the Lausanne University Hospital (CHUV) and the Cantonal Human Research Ethics Committee, can the authors provide the IRB approval number?

3.          Please provide the scale bar in Figure 2b.

4.          In Figure 3b, the amount of total protein staining used as a reference protein for western blotting appears to be different in each lane.

5.          In Figure 5, the adenoviral-mediated overexpression or siRNA-mediated knockdown of CSE is confirmed by Western blotting. However, the efficiency of infection or transfection is not shown or reported. It would be important to show the GFP expression (for adenovirus) or the transfection efficiency (for siRNA) by fluorescence microscopy or flow cytometry to ensure that all cells are infected or transfected.

6.          What was the concentration of DAz-2-Cy5.5 solution?

7.          Why CSE overexpression caused the morphology change in VSMCs?

8.          In Figure 6, the VSMC proliferation is measured by BrdU incorporation. However, this method only reflects DNA synthesis. Would cell death or cell cycle arrest influence the results?

9.          In Figure 6, the VSMC migration is measured by wound healing assay. However, this method is influenced by cell proliferation and does not reflect directional migration. Would it be more relevant to measure cell migration by transwell migration assay or Boyden chamber assay?

10.      The authors have shown that CSE expression is negatively regulated by arterial shear stress in human veins ex vivo. However, they have not investigated whether this regulation is mediated by transcriptional or post-transcriptional mechanisms. Previous studies have suggested that shear stress can affect CSE mRNA stability or translation via microRNAs or RNA-binding proteins. Please briefly discuss it.

11.      The authors should discuss the severe limitations of their approach and give future directions for research in the field of smooth muscle migration in human saphenous vein.

Author Response

Bellow is our point by point answer to the comment of the reviewer.

  1. The authors have provided a comprehensive and novel study on the expression and regulation of H2S-producing enzymes in human vessels and their role in IH. The study is well-designed, executed, and presented, and the results are of high interest and relevance to vascular biology and surgery.

We thank the reviewer for his positive feedback.

  1. The study protocols for organ collection and use were reviewed and approved by the Lausanne University Hospital (CHUV) and the Cantonal Human Research Ethics Committee, can the authors provide the IRB approval number?

The vessel samples used for this study were from discarded tissue collected in the OR during routine surgeries on patients having signed the general consent of the Lausanne university hospital. These samples were further immediately anonymized so that they do not enter in a specific study and require a dedicated consent. As such, we got the approval from the ethics committee to perform these experiments outside of a specific IRB number.

  1. Please provide the scale bar in Figure 2b.

done

  1. In Figure 3b, the amount of total protein staining used as a reference protein for western blotting appears to be different in each lane.

We thank the reviewer for noticing there was a mistake in this figure. As you could see in the supplemental original blot, the total protein loading was not the right one in the main figure. The mistake has been corrected. However, it still looks like different amount of protein was loaded, although protein concentration was measured, and the same amount of protein was loaded in every lane. Working with human vessel samples is tricky as these samples tend to be fibrotic and hard to dissociate. The 7-day culture also change the protein composition of the vessel, leading to different protein loading profiles despite similar amount of protein.

  1. In Figure 5, the adenoviral-mediated overexpression or siRNA-mediated knockdown of CSE is confirmed by Western blotting. However, the efficiency of infection or transfection is not shown or reported. It would be important to show the GFP expression (for adenovirus) or the transfection efficiency (for siRNA) by fluorescence microscopy or flow cytometry to ensure that all cells are infected or transfected.

A new supplemental Figure S4 has been added showing that both RNA silencing and adenoviral infection target most cells. The methods were completed to mention the use of the Adenovirus (CAR) receptor booster to improve infection efficiency in VSMC, which are naturally hard to infect because of low density of the CAR receptor at the surface.

  1. What was the concentration of DAz-2-Cy5.5 solution?

The Daz2-Cy5.5 solution was prepared by mixing 1mM Daz-2, 1mM alkyne Cy5.5, 2mM copper(II)-TBTA, 4mM ascorbic acid and incubating overnight at RT, followed by quenching for 1h with 20mM EDTA. The methods section has been amended for clarity.

  1. Why CSE overexpression caused the morphology change in VSMCs?

Both CSE overexpression and CSE knock-down seem to change VSMC morphology in an opposite way. As mentioned in the discussion, our main finding is that CSE variations reshape VSMC, suggesting an effect on the cytoskeleton and interaction with the ECM. This is consistent with our previous finding that H2S donors inhibit microtubule polymerization in VSMC [1], and the findings that CSE deficiency in mouse VSMC results in increased expressions of β1-integrin and increased migration [2]. In EC, it was recently shown that integrins are extensively sulfhydrated, and that β3 integrin S-sulfhydration promotes adhesion and is required for EC alignment with flow [3]. In this study, they further demonstrated that Cse deficiency in EC leads to overactivation of RhoA, a major hub regulating cell migration and adhesion. Interestingly, β3-integrin is also expressed in VMSC and β3-integrin signaling is instrumental for enhanced VSMC proliferation and migration in vascular disease [4, 5]. We hypothesized that CSE-mediated integrin sulfhydration promotes VSMC adhesion, thereby limiting migration. Further studies are needed to determine the exact role and balance between β1- and β3-integrins and RhoA signaling in the regulation of cell migration by CSE in VSMC.

  1. In Figure 6, the VSMC proliferation is measured by BrdU incorporation. However, this method only reflects DNA synthesis. Would cell death or cell cycle arrest influence the results?

BrdU incorporation allows to mark DNA replication, thus highlighting mitosis events during a given time. Cell cycle arrest is intricately linked to cell proliferation and would influence the results if there was an effect prior to S phase. G2 arrest would lead to successful replication but no mitosis, which would result in low number of cells but high number of BrdU-positive cells. This would be apparent to the observer. Here, we did not observe any difference in cell growth or density upon CSE knock-down not overexpression, and did not push further to measure cell cycle. Cell death would also influence the result, but would also lead to reduced cell density at the end of the experiment. New experiments (Figure 6c-d) have been performed showing that CSE knock-down or overexpression did not influence cell viability.

  1. In Figure 6, the VSMC migration is measured by wound healing assay. However, this method is influenced by cell proliferation and does not reflect directional migration. Would it be more relevant to measure cell migration by transwell migration assay or Boyden chamber assay?

This method is not influenced by cell proliferation as the wound healing assay was performed in presence of 0.5µg/mL mitomycin C to block proliferation. The Boyden chamber assay is indeed a better assay to assess chemotaxis and directional migration. However, our data point at an altered physical ability to migrate rather than an altered response to migratory cues. Due to lack of time, we did not perform a Boyden Chamber assay in this study.

  1. The authors have shown that CSE expression is negatively regulated by arterial shear stress in human veins ex vivo. However, they have not investigated whether this regulation is mediated by transcriptional or post-transcriptional mechanisms. Previous studies have suggested that shear stress can affect CSE mRNA stability or translation via microRNAs or RNA-binding proteins. Please briefly discuss it.

Our data clearly describe a close correlation between CSE mRNA and protein levels and we believe the regulation occurs mainly at the transcriptional level. New data are now included showing that ATF4 is induced upon ex vivo culture (Figure 3b). CTH is a known target of ATF4. Thus, ATF4 overexpression may drive CSE overexpression in response to static ex vivo culture.

KLF2-induced miR-27b expression has also been linked to CTH mRNA down-regulation. Here, we observed that KFL2 expression was severely reduced upon ex vivo static culture, which may contribute to CTH overexpression in this condition. The results have been completed and discussion has been extended to better discuss this point, highlighting possible mechanisms as follows:

Line 478: 

The close correlation between CSE mRNA and protein levels suggest a regulation at the transcriptional level. CTH expression can be induced by stress factor such as endoplasmic reticulum stress, amino acid restriction or oxidative stress via ATF4 [30, 33]. Here, ATF4 was overexpressed in response to static vein culture, which may drive CSE overexpression in that condition. The micro RNA miR-27b, which is highly expressed in EC and involved in angiogenesis [49, 50], has been described to down-regulate CTH expression [21]. However, this mechanism has been described only in EC and miR-27b is down-regulated in EC in pathological condition [21]. miR-27b is under the control of KLF2, a mechanosensitive transcription factor induced by laminar shear stress participating in vascular development [21]. Here, KLF2 expression was severely repressed in static culture, which may result in loss of miR-27b and CTH overexpression. That said, there is no report of miR-27b in VSMC or in the context of IH and further studies are needed to evaluate whether miR-27b in VSMC inhibit CTH expression in relation with flow.

  1. The authors should discuss the severe limitations of their approach and give future directions for research in the field of smooth muscle migration in human saphenous vein.

New limitations and conclusion sections have been added to the discussion.

References

  1. Macabrey D, Longchamp A, MacArthur MR, Lambelet M, Urfer S, Deglise S, et al. Sodium Thiosulfate Acts as a Hydrogen Sulfide Mimetic to Prevent Intimal Hyperplasia Via Inhibition of Tubulin Polymerisation. EBioMedicine (2022) 78:103954. Epub 20220322. doi: 10.1016/j.ebiom.2022.103954.
  2. Yang G, Li H, Tang G, Wu L, Zhao K, Cao Q, et al. Increased Neointimal Formation in Cystathionine Gamma-Lyase Deficient Mice: Role of Hydrogen Sulfide in Alpha5beta1-Integrin and Matrix Metalloproteinase-2 Expression in Smooth Muscle Cells. J Mol Cell Cardiol (2012) 52(3):677-88. doi: 10.1016/j.yjmcc.2011.12.004.
  3. Bibli SI, Hu J, Looso M, Weigert A, Ratiu C, Wittig J, et al. Mapping the Endothelial Cell S-Sulfhydrome Highlights the Crucial Role of Integrin Sulfhydration in Vascular Function. Circulation (2021) 143(9):935-48. Epub 2020/12/15. doi: 10.1161/CIRCULATIONAHA.120.051877.
  4. Misra A, Sheikh AQ, Kumar A, Luo J, Zhang J, Hinton RB, et al. Integrin Beta3 Inhibition Is a Therapeutic Strategy for Supravalvular Aortic Stenosis. J Exp Med (2016) 213(3):451-63. Epub 20160208. doi: 10.1084/jem.20150688.
  5. Slepian MJ, Massia SP, Dehdashti B, Fritz A, Whitesell L. Beta3-Integrins Rather Than Beta1-Integrins Dominate Integrin-Matrix Interactions Involved in Postinjury Smooth Muscle Cell Migration. Circulation (1998) 97(18):1818-27. doi: 10.1161/01.cir.97.18.1818.

Round 2

Reviewer 1 Report

I am still generally supportive to this manuscript and to the scope of the study. However, I would like to kindly ask Authors to carefully reconsider some of my comments:

Revision according to my major comments:

Ad. 1: I appreciate the new data that is valuable and partly addressed the concern. Nevertheless, (mRNA) overexpression of HO-1 or TRX1 is sensitive to oxidative stress but also to many other factors. Thus, these markers reflect possible redox imbalance indirectly, in contrast to other direct parameters such as (but not limited to) e.g. ROS generation, MDA levels, DNA/RNA oxidation (e.g. 8-OHG) and many others. Therefore, if it is not possible to add this sort of data, this issue (the lack of directly measured pro-/antioxidative effects) should be at least listed as a major limitation of the study within the Discussion section.

Ad. 2: Thank you for the clarification. Determination of the downstream target for the mTOR is sufficient and important but S6RP is not the only target and is not regulated solely by mTOR. Therefore, please, revise the appropriate sentence in the Discussion to clearly state that the conclusion (raised by Authors themselves, not by the Reviewer) addressing the activity of mTOR is not based on the direct measurement of mTOR but is reflected by the determination of phosphorylation of S6RP.

Ad. 3: Authors addressed this comment sufficiently.

Revision according to my minor comments:

Ad. 1: Excess of cysteine does not eliminate the activity of CBS which is not residual for L-cysteine metabolism and H2S production. Perhaps, the addition of CBS inhibitor into the reaction mixture would do so. Therefore, based on the current protocol, H2S generation via CSE and CBS activity was measured. Referenced publications (protocols published in Cell) also do not differentiate the CSE and CBS activity.

Ad. 2: It is ok after revision.

Regarding WB: Only one minor issue to be corrected/clarified– possibly the technical minor error during labeling the Figure S5b: original blot image shows that the lower bands are at the level of approx. 40 kDA while in Supplementary Materials (final Figure S5B) it is marked as 12 kDa.

Author Response

Below is our point-by-point answers to the comments raised by the reviewer

Ad. 1: I appreciate the new data that is valuable and partly addressed the concern. Nevertheless, (mRNA) overexpression of HO-1 or TRX1 is sensitive to oxidative stress but also to many other factors. Thus, these markers reflect possible redox imbalance indirectly, in contrast to other direct parameters such as (but not limited to) e.g. ROS generation, MDA levels, DNA/RNA oxidation (e.g. 8-OHG) and many others. Therefore, if it is not possible to add this sort of data, this issue (the lack of directly measured pro-/antioxidative effects) should be at least listed as a major limitation of the study within the Discussion section.

We agree with the reviewer that it would be important to assess redox status in our model. However, the use of ROS or superoxide-sensitive fluorescent probes on frozen section of vessels has been disappointing. Vessels tissue imaging is plagued by autofluorescence, and probes such as ROS-ID® ROS/RNS detection kit (Enzo Life Sciences, Inc.) or H2DCFDA (Thermo) have proven unreliable. The discussion has been modified to state that HO-1 or TRX1 overexpression could be the result of redox imbalance rather than increased H2S production. (line 465).

Ad. 2: Thank you for the clarification. Determination of the downstream target for the mTOR is sufficient and important but S6RP is not the only target and is not regulated solely by mTOR. Therefore, please, revise the appropriate sentence in the Discussion to clearly state that the conclusion (raised by Authors themselves, not by the Reviewer) addressing the activity of mTOR is not based on the direct measurement of mTOR but is reflected by the determination of phosphorylation of S6RP.

The discussion has been modified to address this comment (line 522).

Revision according to my minor comments:

Ad. 1: Excess of cysteine does not eliminate the activity of CBS which is not residual for L-cysteine metabolism and H2S production. Perhaps, the addition of CBS inhibitor into the reaction mixture would do so. Therefore, based on the current protocol, H2S generation via CSE and CBS activity was measured. Referenced publications (protocols published in Cell) also do not differentiate the CSE and CBS activity.

The methods, results and discussion have been modified to indicate that the lead acetate assay measures CSE/CBS-mediated H2S production.

Regarding WB: Only one minor issue to be corrected/clarified– possibly the technical minor error during labeling the Figure S5b: original blot image shows that the lower bands are at the level of approx. 40 kDA while in Supplementary Materials (final Figure S5B) it is marked as 12 kDa.

We thank the reviewer for pointing out this error. The figure S5 has been corrected.